# Data on Orientation to Happiness in Higher Education Institutions from Mexico and El Salvador

**Domingo Villavicencio-Aguilar [1], Edgardo René Chacón-Andrade [2] and Maria Fernanda Durón-Ramos [1,\***

[1]    Instituto Tecnológico de Sonora, Guaymas 85400, Mexico; dvillavicencio@itson.edu.mx
[2]    Universidad Tecnológica de El Salvador, San Salvador 1770, El Salvador; edgardo.chacon@utec.edu.sv
\*    Correspondence: maria.duron@itson.edu.mx; Tel.: +52-622-221-0032

**Abstract:** Happiness-oriented people are vital in every society; this is a construct formed by three different types of happiness: pleasure, meaning, and engagement, and it is considered as an indicator of mental health. This study aims to provide data on the levels of orientation to happiness in higher-education teachers and students. The present paper contains data about the perception of this positive aspect in two Latin American countries, Mexico and El Salvador. Structure instruments to measure the orientation to happiness were administrated to 397 teachers and 260 students. This data descriptor presents descriptive statistics (mean, standard deviation), internal consistency (Cronbach's alpha), and differences (Student's t-test) presented by country, population (teacher/student), and gender of their orientation to happiness and its three dimensions: meaning, pleasure, and engagement. Stepwise-multiple-regression-analysis results are also presented. Results indicated that participants from both countries reported medium–high levels of meaning and engagement happiness; teachers reported higher levels than those of students in these two dimensions. Happiness resulting from pleasure activities was the least reported in general. Males and females presented very similar levels of orientation to happiness. Only the population (teacher/student) showed a predictive relationship with orientation to happiness; however, the model explained a small portion of variance in this variable, which indicated that other factors are more critical when promoting orientation to happiness in higher-education institutions.

**Keywords:** university; happiness; pleasure; meaning; engagement

## 1. Summary

Higher-education institutions often focus their attention to academic engagement [1] and achievement [2], but it is essential that universities also focus on the health of their students [3] and teachers. According to the World Health Organization [4], health is a state that involves mental wellbeing (along with physical and social) that can be promoted by happiness. The present data descriptor includes indicators of mental health through a positive personal trait, orientation to happiness, of both teachers and students in Mexico and El Salvador.

Most people prefer happiness over other essential aspects in their life [5]. Peterson, Park, and Seligman [6] suggested that one of the ways to understand happiness is through a construct called orientation to happiness because it unites three pathways to obtain significant happiness states: pleasure, meaning, and engagement. The first component of orientation to happiness is related to

hedonic wellbeing because it is obtained through pleasure [7]. The second component is compared to eudemonic wellbeing because it is happiness that comes from situations that add meaning to people's lives [7]. Finally, engagement is related to the theory of flow [8] because it is happiness resulting from activities in which the person is absorbed or wholly engaged.

There is empirical evidence that orientation to happiness can enhance students' adaptability at university [9] and engagement in their academic activities [1]. However, data from teachers are not that common in studies involving orientation to happiness. Nevertheless, it is essential to take into account positive personal factors in teachers because they serve as an indicator of health at work, essential according to the Organization of American States (OAS) [10], Pan-American Health Organization (PAHO), and the World Health Organization (WHO) [11]. Global organizations are making efforts to standardize activities towards guaranteeing the wellbeing of workers after the first international standard related to occupational health and safety management (ISO 45001: 2018) was published in March 2018.

Investigations on the perception of teachers (as workers) and students about orientation to happiness are a source for information that supports higher-education institutions in formulating actions to improve mental health in their communities. The majority of studies on higher education present information taken from people living in developed countries; however, this paper presents information from higher-education teachers and students living in two developing countries in Latin America, Mexico and El Salvador.

## 2. Data Description

The project was submitted and approved by the Ethics Committee of the Technological Institute of Sonora (Instituto Tecnológico de Sonora, México) and the Research Council from the Technological University from El Salvador (Universidad Tecnológica de El Salvador). All teachers and students were invited to voluntarily contribute by the research team; participants gave their answers anonymously and individually. The institution from Mexico approved the virtual collection through Google Forms, while the university from El Salvador permitted a written form; therefore, the population from that country answered with a printed copy of the instrument. These answers were introduced (captured) in the same form as that used with the sample from Mexico. The final dataset was downloaded from Google Forms.

We present a data file in a format compatible with the Statistical Package for Social Sciences (SPSS); each line represents a participant (657), while every column is an asked item (question). First, the section of demographic characteristics presented four variables: population (teacher/student), country, gender, and age; second, 18 columns are showing answers from orientation to happiness separated by the three dimensions: pleasure, meaning, and engagement.

The first six items asked questions to measure happiness from pleasure: (a) "Life is too short to postpone the pleasures it can provide"; (b) "I go out of my way to feel euphoric"; (c) "In choosing what to do, I always take into account whether it will be pleasurable"; (d) "I agree with this statement: 'Life is short—eat dessert first'"; (e) "I love to do things that excite my senses"; and (f) "For me, the good life is the pleasurable life".

Items 7–12 present questions related to happiness from meaning: (a) "My life serves a higher purpose"; (b) "In choosing what to do, I always take into account whether it will benefit other people"; (c) "I have a responsibility to make the world a better place"; (d) "My life has lasting meaning", (e) "What I do matters to society"; and (f) "I have spent a lot of time thinking about what life means and how I fit into its big picture".

The last six items contained questions to evaluate happiness from engagement: (a) "Regardless of what I am doing, time passes very quickly"; (b) "I seek out situations that challenge my skills and abilities"; (c) "Whether at work or play, I am usually 'in a zone' and not conscious of myself" (erased after reliability test); (d) "I am always very absorbed in what I do"; (e) "In choosing what to do,

I always take into account whether I can lose myself in it"; and (f) "I am rarely distracted by what is going on around me".

Responses from the orientation-to-happiness scale contained answers using a scale of 1–5, where 1 = completely opposite to me, 2 = different from me, 3 = neutral, 4 = like me, and 5 = very much like me.

### 2.1. Data Reliability and Descriptive Statistics

Table 1 presents the reliability coefficient (Cronbach's alpha) and the descriptive statistics in general from orientation to happiness and its components. Cronbach's alpha shows internal consistency on the scale in general (0.81) and in every dimension (0.66 to 0.76); levels of orientation to happiness are medium–high ($M = 3.85$, $SD = 0.50$), being higher on happiness determined by meaning ($M = 4.05$, $SD = 0.57$), while pleasure was the least-reported ($M = 3.53$, $SD = 0.71$).

**Table 1.** Reliability and descriptive statistics.

| | Cronbach's Alpha | Item Number | Min | Max | *Mean* | *SD* |
|---|---|---|---|---|---|---|
| Orientation to happiness | 0.81 | 17 | 1 | 5 | 3.85 | 0.50 |
| Pleasure | 0.76 | 6 | 1 | 5 | 3.53 | 0.71 |
| Meaning | 0.66 | 6 | 1 | 5 | 4.05 | 0.57 |
| Engagement | 0.66 | 5 | 1 | 5 | 4.01 | 0.61 |

### 2.2. Comparison by Country, Population, and Gender

Descriptive statistics by country (Table 2) indicate that levels of orientation to happiness in general are very similar in Mexico and El Salvador ($t = –0.72$, $p = 0.47$). However, the comparison by dimensions of orientation to happiness showed that people from El Salvador reported statistically higher levels from meaning ($t = –2.77$, $p = 0.01$) and engagement ($t = –2.36$, $p = 0.02$), while Mexicans were higher on pleasure happiness ($t = 2.35$, $p = 0.02$).

**Table 2.** Comparison of orientation to happiness by country.

| | | **México** | | **El Salvador** | | | | |
|---|---|---|---|---|---|---|---|---|
| | Range | Mean | SD | Mean | SD | t | df | P |
| Orientation to happiness | 1–5 | 3.84 | 0.49 | 3.87 | 0.52 | −0.72 | 650.64 | 0.47 |
| Pleasure | 1–5 | 3.60 | 0.64 | 3.47 | 0.75 | 2.35 | 654.37 | 0.02 |
| Meaning | 1–5 | 3.98 | 0.60 | 4.10 | 0.53 | −2.77 | 615.91 | 0.01 |
| Engagement | 1–5 | 3.95 | 0.64 | 4.06 | 0.58 | −2.36 | 626.46 | 0.02 |

Table 3 compares values of orientation to happiness by population (teacher/ student); teachers presented higher levels in general ($t = −2.74$, $p = 0.01$), and at the dimensions of meaning ($t = −4.93$, $p = 0.00$) and engagement ($t = −6.39$, $p = 0.00$); students, on the other hand, presented higher levels in the pleasure dimension ($t = 4.16$, $p = 0.00$).

**Table 3.** Comparison of orientation to happiness by population (teacher/student).

| | | **Teachers** | | **Students** | | | | |
|---|---|---|---|---|---|---|---|---|
| | Range | Mean | SD | Mean | SD | t | df | P |
| Orientation to happiness | 1–5 | 3.90 | 0.48 | 3.79 | 0.53 | −2.74 | 520.59 | 0.01 |
| Pleasure | 1–5 | 3.44 | 0.69 | 3.67 | 0.72 | 4.16 | 533.66 | 0.00 |
| Meaning | 1–5 | 4.14 | 0.52 | 3.91 | 0.62 | −4.93 | 489.80 | 0.00 |
| Engagement | 1–5 | 4.13 | 0.56 | 3.82 | 0.63 | −6.39 | 509.10 | 0.00 |

Comparisons by gender are presented in Table 4. Values in general, separated by the three dimensions, were similar between men and women because p-values were between 0.40 and 0.79.

**Table 4.** Comparison of orientation to happiness by gender.

| | | Males | | Females | | | | |
|---|---|---|---|---|---|---|---|---|
| | Range | Mean | SD | Mean | SD | t | df | P |
| Orientation to happiness | 1–5 | 3.84 | 0.50 | 3.84 | 0.50 | −0.84 | 651.87 | 0.40 |
| Pleasure | 1–5 | 3.52 | 0.72 | 3.52 | 0.72 | −0.27 | 647.80 | 0.79 |
| Meaning | 1–5 | 4.03 | 0.60 | 4.03 | 0.60 | −0.58 | 638.02 | 0.56 |
| Engagement | 1–5 | 3.99 | 0.59 | 3.99 | 0.59 | −0.61 | 654.81 | 0.55 |

The influence of country, population (teacher/student), and gender variables on overall orientation to happiness was tested through stepwise-regression analysis; these three predictors explained variance in a very small proportion, $R^2 = 0.012$, $F (1655) = 7.748$, $p < 0.01$. Only the fact of being a teacher or student showed statistical significance when predicting orientation to happiness ($\beta = 0.108$, $p = 0.006$); country and gender were excluded from the model (country: $\beta = 0.002$, $p = 0.970$; gender: $\beta = 0.030$, $p = 0.436$).

## 3. Methods

### 3.1. Participants

Teacher sample: 397 higher-education teachers from Mexico and El Salvador gave answers. Of this population, 53% were women and 47% men; ages ranged from 21 to 80 (M = 43.20, SD = 10.85); 63% of the teachers that participated were from Salvador and 37% from Mexico.

Student sample: 260 university students from two higher-education institutions participated in this research. Gender was equal, with 50% for men and women; the age range was from 18 to 53 (M = 23.27, SD = 6.75); and 62% of the students were from Mexico, while 38% were from El Salvador.

### 3.2. Instruments

The orientation-to-happiness scale [6] was used in its Spanish version [12]. It contains 18 items divided into the three types of happiness: pleasure, meaning, and engagement, with 6 for each one. The participants could respond using a scale that was from 1 = completely opposite to me, to 5 = very much like me. This is broadly described in Section 2.

### 3.3. Procedure

The sample size was estimated with a 5% margin of error and 95% confidence (see Table 5). Participants voluntarily gave their answers in a virtual format for Mexico and in a written paper for El Salvador (as described in Section 2). Data analysis started with a reliability test using Cronbach's alpha. In this procedure, a question from engagement had to be left out because it contained a metaphor ("in the zone") that US speakers often use but was confusing for the population living in Mexico and El Salvador; therefore, internal consistency of this dimension was altered with this item.

After reliability was confirmed, four variables were made by calculating the mean for every dimension and the total scale; this gave the result of four indicators: orientation to happiness (total scale for 17 items), pleasure (6 items), meaning (6 items), and engagement (5 items). Descriptive statistics were obtained, along with the Student's t-test to compare means by country, population (teacher/student), and gender. Finally, stepwise-multiple-regression analysis was performed between orientation to happiness (dependent variable) and the three demographic variables that were used to compare as independent variables (country, population, and gender). Assumptions were tested by examining normal probability plots; no violation of normality was detected.

**Table 5.** The sample size.

| Country | Population | Total | Sample |
|---------|-----------|-------|--------|
| Mexico | Teachers | 237 | 146 |
| | Students * | 305 | 161 |
| El Salvador | Teachers | 570 | 251 |
| | Students * | 175 | 99 |

* Conventional student (face-to-face).

**Author Contributions:** D.V.-A., E.R.C.-A., and M.F.D.-R. conceived and designed the study. D.V.-A. and E.R.C.-A. collected the data. M.F.D.-R. analyzed the data and wrote an initial draft on the basis of the results. D.V.-A. and E.R.C.-A. critically revised the draft manuscript and made important changes to the content. All authors have read and agreed to the published version of the manuscript.

**Funding:** Publication funded with resources from PFCE 2019 (Programa para el Fortalecimiento de la Calidad Educativa/Program for Strengthening Educational Quality).

**Conflicts of Interest:** The authors declare no conflict of interest.

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
