# Peer review of "Data on Orientation to Happiness in Higher Education Institutions from Mexico and El Salvador"

_data_

Round 1
Reviewer 1 Report
As a general note, the text is well conducted, and it's easy to understand by the reader.
Author Response
Reviewer comment: As a general note, the text is well conducted, and it's easy to understand by the reader.
The authors thank the reviewer for the effort and time spent in our manuscript. We are pleased to note that you find it suitable and well-conducted.

Reviewer 2 Report
Please bear in mind that this paper is reviewed according to criteria for Data Descriptor.
The paper presents data on quality of life and well-being from teachers and students from two developing countries. The data is analyzed by performing statistical test (t-test) to compare different groups of respondents. However, more advanced methods (such as PCA or MLR) could provide interesting results on the influences of each dimension on overall orientation to happiness. To make data more useful for future research the authors should provide information on population size for the countries analyzed, margin of error and confidence interval of the sample.
Author Response
The authors thank for the time spent in reviewing this manuscript; following up these recommendations, the manuscript has been improved. Responses by authors are given in bold typeface and reviewer’s comments in Italics.
The paper presents data on quality of life and well-being from teachers and students from two developing countries.
Point 1: The data is analyzed by performing statistical test (t-test) to compare different groups of respondents. However, more advanced methods (such as PCA or MLR) could provide interesting results on the influences of each dimension on overall orientation to happiness.
Response 1: A stepwise multiple regression analysis was performed, this information was added on the abstract (see page 1, lines 18-19, and lines 22 to 25), procedure (see page 4, lines 140 to 144) and results (see page 4, lines 110 to 115).
Point 2: To make data more useful for future research the authors should provide information on population size for the countries analyzed, margin of error and confidence interval of the sample.
Response 1: The sample size was estimated with a 5 % margin of error and 95% confidence; this information is presented on page 4, line 134. Given the fact that we used four different populations (Teachers from Mexico, Teachers from El Salvador, Students from Mexico, and students from El Salvador), population size information is included in a table as a supplementary material.

This manuscript is a resubmission of an earlier submission. The following is a list of the peer review reports and author responses from that submission.
Round 1
Reviewer 1 Report
Thank you for inviting me to review the manuscript: Data on quality of life and orientation to happiness in higher education teachers from Mexico and El Salvador. Villavicencio-Aguilar et al. have reported two positive aspects perceived by higher education teachers. I have a few concerns about the present manuscript:
The abstract is incomplete: The authors did not describe the background, aims, and results of the study. Also, there is repeated information about statistical analysis. Summary: The authors indicated studies in the introduction, but they did not inform the bibliographic references. In addition, there is not a robust background to support the idea of the study. The authors did not inform the words or phrases of QOL and OTH abbreviations. How the data were analyzed after they were collected? The results were indicated in the tables, but they were not described in the text. There is no discussion to support the importance of the authors’ data. Although the data are original, and the study presents the informed sources, the methodology is not well established. For example, the study has 397 participants, but there is no detailed statistical analysis. The authors reported frequencies and standard deviations, only. The authors performed the statistical analysis by Cronbach’s test. But they didn’t inform in the methods section.
Reviewer 2 Report
The paper presents data on quality of life and well-being and could provide useful information for further analysis of these topics. Moreover, contrary to the vast majority of papers, the data presented focus on subjects from developing countries and could be used (for example) for comparative analyses between devoloped and developing countries.
However, even though most probably the data is intended to be used in social science studies and to provide insights on various trends, it would be helpful to have more information on participants and population. The authors should provide information on population size for the countries analyzed, margin of error and confidence interval of the sample.